# Anti-Inflammatory Activity of Geraniol Isolated from Lemon Grass on Ox-LDL-Stimulated Endothelial Cells by Upregulation of Heme Oxygenase-1 via PI3K/Akt and Nrf-2 Signaling Pathways

**DOI:** 10.3390/nu14224817

**Published:** 2022-11-14

**Authors:** Rebai Ben Ammar, Maged Elsayed Mohamed, Manal Alfwuaires, Sarah Abdulaziz Alamer, Mohammad Bani Ismail, Vishnu Priya Veeraraghavan, Ashok Kumar Sekar, Riadh Ksouri, Peramaiyan Rajendran

**Affiliations:** 1Department of Biological Sciences, College of Science, King Faisal University, Al-Ahsa 31982, Saudi Arabia; malfwuaires@kfu.edu.sa (M.A.); salamer@kfu.edu.sa (S.A.A.); 2Laboratory of Aromatic and Medicinal Plants, Center of Biotechnology of Borj-Cedria, Technopole of Borj-Cedria, P.O. Box 901, Hammam-Lif 2050, Tunisia; ksouri.riadh@gmail.com; 3Department of Pharmaceutical Sciences, College of Clinical Pharmacy, King Faisal University, Al-Ahsa 31982, Saudi Arabia; memohamed@kfu.edu.sa; 4Department of Pharmacognosy, Faculty of Pharmacy, University of Zagazig, Zagazig 44519, Egypt; 5Department of Basic Medical Sciences, School of Medicine, Aqaba Medical Sciences University, Aqaba 11191, Jordan; m.bani0331@gmail.com; 6Centre of Molecular Medicine and Diagnostics (COMManD), Department of Biochemistry, Saveetha Dental College & Hospitals, Saveetha Institute of Medical and Technical Sciences, Saveetha University, Chennai 600077, Tamil Nadu, India; vishnupriya@saveetha.com; 7Centre for Biotechnology, Anna University, Chennai 600025, Tamil Nadu, India; ashokkumar@annauniv.edu

**Keywords:** anti inflammation, Ox-LDL, geraniol, oxidative stress

## Abstract

Among the world’s leading causes of cardiovascular disease, atherosclerosis is a chronic inflammatory disorder that affects the arteries. Both vasodilation and vasoconstriction, low levels of nitric oxide and high levels of reactive oxygen species and pro-inflammatory factors characterize dysfunctional blood vessels. Hypertension, and atherosclerosis, all start with this dysfunction. Geraniol, a compound of acyclic monoterpene alcohol, found in plants such as geranium, lemongrass and rose, is a primary constituent of essential oils. It shows a variety of pharmacological properties. This study aimed to investigate the impact of geraniol on Ox-LDL-induced stress and inflammation in human umbilical vein endothelial cells. In this study, HUVECs were treated with Ox-LDL or geraniol at different dose concentrations. MTT assay, Western blot, ROS generation and DNA fragmentation were used to evaluate geraniol’s effects on Ox-LDL-induced HUVECs inflammation. The results show that geraniol pre-incubation ameliorates Ox-LDL-mediated HUVECs cytotoxicity and DNA fragmentation. The geraniol inhibited the production of pro-inflammatory cytokines by Ox-LDL, including TNF-α, IL-6 and IL-1β. In Ox-LDL-stimulated HUVECs, geraniol suppresses the nuclear translocation and activity of NF-ᴋB as well as phosphorylation of IkBα. Moreover, geraniol activated the PI3K/AKT/NRF2 pathway in HUVECs, resulting in an increase in the expression of HO-1. Taking our data together, we can conclude that, in HUVECs, geraniol inhibits Ox-LDL-induced inflammation and oxidative stress by targeting PI3/AKT/NRF2.

## 1. Introduction

The leading cause of morbidity and mortality in many countries is cardiovascular disease (CVD). Heart disease is mostly caused by atherosclerosis (AS). In the past decade, AS has caused an increase in coronary heart disease morbidity [1]. Every year, almost 400 out of every 100,000 people die from the disease in Asia [1,2]. In addition, AS is considered the primary cause of non-infectious disease-related mortality worldwide. It’s widely accepted that endothelial activation and/or dysfunction are the earliest signs of atherosclerosis [3]. Ox-LDL induces the expression of pro-inflammatory genes, leading to the recruitment of monocytes into the vessel wall and vascular dysfunction [4,5]. Endothelial cells are damaged by free radicals and nitric oxide synthase gene expression and activity are impaired. Ox-LDL triggers adhesion molecules, including ICAM-1 and VCAM-1. Bifurcations and curvatures cause these adhesion molecules to be strongly expressed. By activating VCAM-1, monocytes get rolled along and adhere to the arterial wall, where they can invade the intima, causing foam cells and increasing intima-media thickness, inflammation and oxidative stress [6,7,8,9].

As adhesion molecules accumulate, inflammatory cells adhere to the endothelium, leading to AS [10]. Inflammation and oxidative stress are both fundamental mechanisms associated with various pathologies. ROS are a major target of oxidant stress and play a critical role in several diseases and disorders of the vascular system. Ox-LDL has been shown to cause intracellular ROS production in endothelial cells and alter their signaling [11]. Ox-LDL suppresses NF-ᴋB and stops endothelial cells from committing apoptosis, while other studies suggest it induces NF-ᴋB activation and the expression of pro-inflammatory genes [9,12,13].

The cytoprotective enzyme HO-1 controls inflammation and apoptosis [14]. Studies have shown that activating Nrf2 can boost HO-1 activation [15,16,17,18]. When Nrf2 is not working, HO-1 expression is inhibited, and oxidative stress is exacerbated. Due to its repressor, Keap1, Nrf2 is inactive in the cytoplasm under physiological conditions. During oxidative stress, in the nucleus, Nrf2 is released from the Keap1–Nrf2 complex. A number of studies have shown that activating the PI3K/AKT signaling pathway can facilitate nuclear translocation of Nrf2 [9,19,20]. Inflammation and induced Ox-LDL may be prevented by activating HO-1 and modulating it. The Nrf2–HO-1 pathway is activated by stimulated antioxidants. It has been found that some natural compounds trigger Nrf2 nuclear translocation and protect endothelium dysfunction [21,22,23].

Geraniol (GNL) is a monoterpene (Figure 1). There are lots of essential oils extracted from lemongrass, rose, lavender and other aromatic plants. It exerts a range of pharmacological effects, including antibacterial, anti-inflammatory, antioxidant and neuroprotective [24,25,26,27,28,29]. As an antimicrobial agent, geraniol effectively prevented dysbiosis associated with colitis in mice. Geraniol also reduced apoptotic markers and enhanced dopaminergic neurons’ ability to survive free-radical damage [28].

We hypothesize that GNL has cognitive enhancement effects on a D-gal-mediated ageing model based on our previous findings. GNL has been shown to reduce Ox-LDL-induced endothelial damage for the first time. Hopefully, our research can shed some light on GNL’s potential clinical application for Ox-LDL-related cardiovascular diseases. 

## 2. Materials and Methods

### 2.1. Reagents

HUVECs were obtained from the Chinese Academy of Sciences in Shanghai. Ox-LDL was supplied by UnionBio Technology (China). 2,7-Dichlorodihydrofluorescein diacetate (DCFH2-DA) and methyl thiazolyl tetrazolium (MTT) were from Sigma, and DMEM from Gibco BRL/Invitrogen. All chemicals used were analytical grade.

#### 2.1.1. Plant Materials

In April 2020, wild lemongrass was obtained from native farms in Al-Asha, Eastern Province, K.S.A. Taxonomists at King Saud University identified the plant. We deposited the samples in the herbarium at King Faisal University, Saudi Arabia, with receipt no. P1906.

#### 2.1.2. Isolation of Lemongrass Essential Oil

A coarse powder was made by pulverizing the dried aerial parts of *Cymbopogon commutatus* (2000 g) and hydrodistilling it [30,31].

#### 2.1.3. Lemongrass Essential Oil Analysis

According to Perveen et al., lemongrass vital oil was examined using gas chromatography–mass spectrometry (GC–MS) [30], with some modifications. Younis et al. has described the gas chromatography/flame ionization analysis (GC/FID) (2021) [31].

#### 2.1.4. Separation of GNL from the Lemon Grass Vital Oil

A column chromatography method was used to remove GNL from the oil mixtures by Savira et al. (2019) [32], with some modification. The different isolation steps were previously described by Younis et al. (2021) [31].

### 2.2. Culture and Treatment

We plated HUVECs in DMEM medium and cultured them at 37 °C with 5% CO_2_. Serum-free medium was used to suspend GNL and Ox-LDL. Each group of cells were seeded onto 6-well plates, and then GNL, Ox-LDL, or a mixture of GNL, Ox-LDL, and a mixture of the two was added over 24 h in triplicate.

### 2.3. MTT Assay

An MTT kit was used, as per the manufacturer’s instructions, to measure the cytotoxicity of GNL on HUVECs. In 96-well plates (5 × 10^3^ cells per well), the 15-passage HUVECs were incubated for 24 h with different concentrations (0, 25, 50, 100 µM) of GNL and/or Ox-LDL for 24 h. We measured the absorbance of each well at 570 nm with a microplate reader (Winoosky, VT, USA). We used untreated cells to test cell viability (set to 100%).

### 2.4. Analysis the Level of IL-1β, TNF-α, and ICAM-1 in Culture Supernatant 

We cultured HUVECs in a 12-well plate with about 6.5 × 10^5^ cells/well. Pretreatment with GNL (0–100 µmol/mL, 2 h) followed by OxLDL treatment (100 µg/mL, 72 h), culture supernatants were collected followed by IL-1β, TNF-α, and ICAM-1 were quantified using their respective ELISA kits (Appendix A) [33,34].

### 2.5. Western Blot

A radioimmunoprecipitation assay was employed to isolate the whole-cell proteins from the HUVECs. Using the Nuclear and Cytoplasmic Protein Extraction Kits, we isolated the nuclear and cytoplasmic proteins. Afterwards, lysates on ice were treated with ultra-wave sonication, followed by centrifugation at 12,000× *g* rpm at 4 °C for 30 min. BCA protein assays were also used to measure the protein concentration. Each sample (20 µg) was separated by sodium dodecylsulfate-polyacrylamide gel electrophoresis (SDS PAGE), and then transferred to a polyvinylidene fluoride membrane (PVDF) (Bio-Rad Laboratories, Hercules, CA, USA). Membranes were blocked for 2 h with 5% nonfat milk, and then probed with primary antibodies. Dilution is shown in Appendix A. Afterwards, PVDF membranes were hatched with secondary antibodies for 1 h, and a chemiluminescence substrate was used to develop them (Pierce Biotechnology, Rockford, IL, USA).

### 2.6. Measurement of ROS Generation

Through fluorescence microscopy, the cell-permeable fluorogenic test detected intracellular ROS. We performed a ROS generation assay using DCFH2-DA, as reported previously.

### 2.7. RT-PCR

As per the manufacturer’s instructions, we used a PrimeScript RT reagent kit (Takara Bio, Shiga, Japan) to convert the RNA to cDNA after washing the cells with PBS and applying TRIzol reagent (Invitrogen, Carlsbad, CA, USA) to isolate the HUVECs’ RNA. Appendix A shows the primer sequences. We then used SYBR Green (Applied Biosystems, Foster City, CA, USA) and ViiA-7 Applied Biosystem (Carlsbad, CA, USA) to do real-time qPCR. The expression of mRNA was standardized to the β-actin housekeeping gene. We compared the expression of mRNA between groups by comparing the 2^ΔΔ^ Ct values to the non-treated samples.

### 2.8. DNA Fragmentation

HUVECs were treated with Ox-LDL and/or GNL and DNA fragmentation was measured with the Cell Death Detection ELISA PLUS kit (Roche Applied Science, Branford, CT, USA), as per the manufacturer’s instructions.

### 2.9. Statistics

We performed a one-way ANOVA. Subsequently, we used Tukey’s post-hoc test, and *p* < 0.05 was considered statistically significant.

## 3. Results

### 3.1. GNL Ameliorated Ox-LDL-Induced Cell Death in HUVECs

In this study, we aimed to determine the cytotoxicity and viability of HUVECs treated with GNL as well as the effect of GNL on Ox-LDL-induced cell death in HUVECs. MTT assays revealed that GNL alone did not exhibit significant cytotoxicity at a dose of 25–100 µM/mL. Besides, there is no statistical significance between the different dose concentrations (Figure 2A). Figure 2B shows the morphology appearance of HUVECs treated or not with GNL at 50 or 100 µM/mL during 24 h. Cells are normal and healthy and only a few are detached in the medium. On the other hand, Ox-LDL (100 µg/mL) significantly (*p* < 0.05) inhibited the proliferation of HUVECs. In contrast, GNL protected cells from Ox-LDL-induced cell death (Figure 2C) (*p* < 0.05). The HUVECs’ morphology was changed by Ox-LDL (100 g/mL). These cells showed the characteristics of apoptotic cells, such as shrinkage of the cell body and membrane blebbing (Figure 2D), while the GNL with Ox-LDL-treated cells had a normal architecture. At a higher concentration (100 µM/mL) of GNL, this effect was significantly and remarkably suppressed. Overall, the data above revealed that GNL might be a potent inhibitor of Ox-LDL-induced endothelial dysfunction. 

### 3.2. GNL Reduced Ox-LDL-Induced Cytokine Production

Western blot was used to measure the effect of GNL on Ox-LDL-induced inflammation in HUVECs. Ox-LDL-treated cells had a significantly elevated level of TNF-α, IL-6, IL-1β and TGF-β (*p* < 0.05). The pre-incubation with GNL significantly (*p* < 0.05) reduced the expression of pro-inflammatory cytokines TNF-α, IL-6, IL-1β and TGF-β in HUVECs compared with the Ox-LDL alone group (Figure 3). When cells were treated with 100 µM GNL, cytokine production was more inhibited than when they were treated with 50 µM. Similar changes in protein abundance IL-6, TNF- and IL-1 were found in the HUVEC culture supernatants during the same study (Appendix A).

### 3.3. Effect of GNL on Endothelial Dysfunction

By accelerating endothelial cell dysfunction, Ox-LDL plays a major role in GNL lesion formation. The proinflammatory vascular adhesion molecules VCAM-1 and ICAM-1 are biomarkers of endothelial dysfunction [35].

VCAM-1 and ICAM-1 were analyzed by Western blot in HUVECs. When Ox-LDL was added, VCAM-1 and ICAM-1 were increased 4.4–3.6-fold (*p* < 0.05), whereas pre-treatment of GNL (50 and 100 µM/mL) lowered their expression level by 2.4–1.5-fold (Figure 3). When cells were treated with 100 µM GNL, there was more inhibition than when they were treated with 50 µM.

### 3.4. Effect of GNL on the Ox-LDL-Induced mRNA Level of TNF-α, IL-6 and ICAM-1

Considering the role of GNL in endothelial dysfunction, we analyzed whether GNL was involved in the inhibition inflammatory pathway activation in Ox-LDL-supplemented cells. qRT-PCR showed significant downregulation of TNF-α, ICAM-1 and IL-6 mRNA in GNL-pretreated cells (50 and 100 µM) (*p* < 0.05), compared to Ox-LDL-treated cells (Figure 4A–C). When cells were treated with 100 µM GNL, there was more inhibition than when they were treated with 50 µM. Inflammatory responses to Ox-LDL were downregulated by GNL, according to these data.

### 3.5. GNL Decreases the Ox-LDL Effect on Free Cholesterol and Total Cholesterol in HUVECs

Plaque formation starts with a lesion in the endothelial layer, which allows LDL to move from the blood into the intima and become Ox-LDL [36]. The concentration of cholesterol in the blood has been recognized as an indicator for plaque development, where higher LDL and lower HDL levels indicate a higher chance of plaque formation [37]. As a next step, we checked whether or not GNL had an effect on improving the increase in free cholesterol caused by Ox-LDL in the HUVECs. Ox-LDL increased (*p* < 0.05) the free cholesterol level in the HUVECs, but because of the induction of 100 µM doses of GNL, the high level decreased (Figure 4D) when compared to GNL 50 µM. Treatment with GNL (50 and 100 µM) also reduced the total cholesterol (Figure 4E) despite Ox-LDL increasing the total cholesterol (*p* < 0.05). When cells were treated with 100 µM GNL, there was more effect than when they were treated with 50 µM.

### 3.6. GNL Could Inhibit FABP4 Protein Activation in Ox-LDL-Induced HUVECs

FABP4, a member of the fatty acid-binding proteins family, plays a crucial role in lipid metabolism and inflammation. FABP4 may be a useful biomarker for patients with stable peripheral vascular disease at risk of major cardiovascular problems, according to previous studies [38,39]. Further, FABP4-deficient macrophages in atherosclerosis show a significantly declined appearance of inflammatory cytokines. For this reason, next we determined the activation of FABP4 in Ox-LDL and/or GNL-treated HUVECs. An increase in FABP4 expression was seen in endothelial cells stimulated with 100 µg/mL Ox-LDL compared to the control group (Figure 4F). In contrast, GNL downregulates incremental FABP4 concentration-dependently. This result indicates that GNL could reduce the expression of the FABP4 gene in Ox-LDL-induced vascular dysfunctions.

### 3.7. GNL Inhibits ROS Formation on Ox-LDL-Induced Endothelial Cells

HUVECs were labeled with H2DCFDA in order to clarify the inhibitory effect of GNL on Ox-LDL-induced oxidative stress. Fluorescence microscopy was used to visualize ROS production (Figure 5A). Ox-LDL induced intracellular ROS levels significantly (*p* < 0.05) in Figure 5A compared to the untreated cells. On the other hand, pretreatment with GNL (50 and 100 µM) significantly reduced Ox-LDL-induced ROS levels (*p* < 0.05). When cells were treated with 100 µM GNL, there was more inhibition than when they were treated with 50 µM.

### 3.8. GNL Inhibts MDA Formation in Ox-LDL-Treated Oxidative Stress

Figure 5B shows the levels of MDA in the control and experimental HUVECs. Cells that were treated with Ox-LDL had higher levels of MDA (*p* < 0.05) when compared to cells that were not treated. Although MDA was inhibited in GNL- (50 and 100 µM) and Ox-LDL-exposed cells (*p* < 0.05), when cells were treated with 100 µM GNL, there was more inhibition than when they were treated with 50 µM GNL—alone treated cells did not show any difference from the controls. The results of this study suggest that GNL may inhibit MDA formation in HUVECs with Ox-LDL-induced vascular dysfunction.

### 3.9. Effect of GNL on Antioxidant Enzymes

It is well known that mammals, including endothelial cells, have developed a system that defends them by enhancing antioxidant enzymes, so oxidative stress injury can be reduced. Compared to the control group, SOD and CAT activity decreased significantly in the groups that received Ox-LDL (*p* < 0.05). There was an increased amount of Ox-LDL in the GNL (50 and 100 µM) when HUVECs were treated with Ox-LDL (*p* < 0.05) (Figure 5B). When cells were treated with 100 µM GNL, there was more activity than when they were treated with 50 µM. Based on the above results, it can be concluded that GNL plays a suppressive role in the Ox-LDL-induced oxidative stress of cells in the endothelium.

### 3.10. Effect of GNL on Ox-LDL-Induced NF-κB p65 Expression

Cytokines are involved in the phosphorylation of NF-ᴋB and the degradation of IᴋB-α. It is necessary for NF-ᴋB to be activated in order for p65 subunit to translocate from the cytoplasm to the nucleus, where it acts as a transcription factor for genes related to cell proliferation. Therefore, we examined the effect of GNL on NF-ᴋB activation and translocation in this study. In cells pretreated with GNL (50 and 100 µM), phosphorylation of IᴋB-α in the total extract was inhibited (*p* < 0.05). Ox-LDL increased NF-ᴋB p65 translocation into the nucleus, whereas GNL (50 and 100 µM) inhibited this translocation in a dose-dependent manner (Figure 6A). When cells were treated with 100 µM GNL, there was more inhibition than when they were treated with 50 µM.

### 3.11. GNL Stimulated Nrf2, HO-1 and NQO1 Expression in Ox-LDL-Treated Endothelial Cells

In earlier studies, Nrf2 played a role in anti-inflammatory processes such as inflammatory cell recruitment and gene expression regulation. In addition to being an antioxidant pathway, Keap-1/NRF2/ARE regulates anti-inflammatory gene expression, which prevents inflammation from progressing [17,40,41]. We measured the levels of several antioxidant proteins. In this study, Western blots showed that Ox-LDL-treated cells showed significantly (*p* < 0.05) reduced expression of antioxidants proteins, whereas the GNL (50 and 100 µM) treatment significantly increased (*p* < 0.05) the expression of Nrf2, HO-1, NQO-1 and γ-GCLC in the nucleus (Figure 6B). At varying dose points, we measured the fold over basal levels of the antioxidant protein expression. When cells were treated with 100 µM GNL, there was more stimulation than when they were treated with 50 µM. There was a differential but substantial effect of GNL on nuclear Nrf2, total Nrf2, HO-1, γ-GCLC and NQO-1 expression patterns.

### 3.12. GNL Upregulates PI3K/AKT Phosphorylation in Ox-LDL-Induced HUVEcs

Many studies suggest ROS activate apoptosis via the PI3/AKT pathway. There is evidence that oxidative stress decreases PI3/AKT phosphorylation and inactivates various transcription factors, such as Nrf2 [42,43,44]. To determine whether Ox-LDL does or does not affect PI3/AKT phosphorylation expression, HUVECs were induced and exposed to Ox-LDL for various dose periods (Figure 7A). After Ox-LDL exposure for the indicated dose, Western blotting revealed a significant decrease in PI3/AKT expression (*p* < 0.05). When exposed to GNL (0–100 µM), HUVECs showed a dose-dependent increase in *p*-PI3K and *p*-AKT expression (*p* < 0.05). When cells were treated with 100 µM GNL, there was more phosphorylation than when they were treated with 50 µM. AKT and PI3K, however, remained relatively unchanged. As a result, the level of pPI3K and pAKT cells increased significantly after GNL treatment. GNL seems to protect against Ox-LDL-induced oxidative stress through a signaling pathway such as PI3K/AKT.

### 3.13. GNLActivates PI3K/AKT Signaling to Regulate Nrf2 in Ox-LDL-Induced Endothelial Cells

In order to determine whether Nrf2 signals contributed to the upregulation of pAKT, Western blot analysis was performed. The aim of this experiment was to see if GNL and Ox-LDL activated Nrf2 signaling via PI3K/AKT. Figure 7B shows that cells treated with Ox-LDL and GNL upregulated pAKT and Nrf2. These levels were significantly downregulated by LY294002-treated cells and GNL + Ox-LDL. It was concluded that GNL may be dependent on PI3K/AKT–Nrf2 signaling in HUVECs.

### 3.14. GNL Upregulates HO-1 Activation through the Nrf2 Pathway

As a result of these findings, we examined the effect of GNL on Ox-LDL-induced oxidative stress when Ho-1 inhibitors were present. We studied the kind of impact Snpp had on the expression of the aforementioned protein signals in order to find out whether the GNL pathway activated by GNL played a role in ensuring HUVECs survival. Ox-LDL and/or GNL increased protein expression in comparison to Ox-LDL alone, though the Snpp result largely disputes this finding (Figure 7C). Snpp also stopped the GNL-induced rise in HO-1 protein expression. 

### 3.15. GNL Inhibits Apoptosis in HUVECs Line

Apoptosis is characterized by DNA fragmentation. Using an ELISA kit, we tested DNA fragmentation for cell death [45]. Ox-LDL-treated cells saw a significant increase in DNA fragmentation, while GNL-treated cells saw a marked decrease in DNA fragmentation (Figure 7D) following treatment, in comparison to the cells treated with a combination of GNL and Ox-LDL. When cells were treated with 100 µM GNL, there was more inhibition than when they were treated with 50 µM. Still, GNL inhibited apoptosis in vascular endothelial cells induced by Ox-LDL-induced oxidative stress.

## 4. Discussion

Hypertension, hyperglycemia, hyperlipidemia, smoking and even drinking can cause AS. Vascular endothelial injury, inflammation, oxidative stress and apoptosis are believed to be the main causes of AS, despite not being fully understood [10]. LDL cholesterol accumulates under the inner membrane when endothelial cells are damaged. An oxidized and modified LDL-C can be engulfed by macrophages and form foam cells, triggering AS, thrombosis and heart attacks. Additionally, when the body is stimulated by oxidative stress factors, such as hydrogen peroxide, low-density lipoprotein (LDL) and proinflammatory factors, it produces a lot of reactive oxygen species (ROS), which can upset the oxidant–antioxidant balance [46,47,48]. Endothelial cells are overactivated by oxidative stress, which reduces their repair ability [49]. To prevent and treat AS, Ox-LDL should be reduced to prevent oxidative damage to endothelial cells. Blood vessels are kept dynamic by the proliferation and apoptosis of endothelial cells. Apoptosis of endothelial cells is a symptom of endothelial dysfunction in AS plaques [3,50,51]. An MTT assay was used to measure the viability of HUVECs, and Ox-LDL significantly decreased their viability. GNL reduced Ox-LDL-induced cell damage in HUVECs in a concentration-dependent manner. AS is mostly caused by inflammation of the vascular endothelium. Inflammatory responses in the vascular endothelium involve rolling, adhesion, migration and accumulation of white blood cells [50,52]. Various inflammatory factors and adhesion molecules can be produced when endothelial cells are stimulated by external factors. Inflammation will result from the adhesion of white blood cells to endothelial cells. Xiaozhen et al. [53] found that treatment with Ox-LDL increased IL-6, TNF-α, IL-β1 VCAM1, ICAM1 and TGF-β. Studies showed that GNL reduces inflammatory and adhesion factors in AS [54,55]. Ox-LDL significantly upregulated TNF-α, IL-6, VCAM1, ICAM1 and IL-1β expression in endothelial cells, while GNL treatment significantly (*p* < 0.05) downregulated these expressions, which agrees with previous studies. Ox-LDL induced an inflammatory response in HUVECs, and GNL protected them. 

Oxidative stress causes abnormal apoptosis and inflammation when ROS levels exceed the body’s antioxidant ability. Dyslipidemia and metabolic syndrome increase ROS [56,57]. Antioxidases protect the body from oxidative damage. SOD and CAT expressions are downregulated by Ox-LDL, while SOD and CAT expressions are upregulated by GNL. GNL protects HUVECs from oxidative stress induced by Ox-LDL. Mechanisms may involve apoptosis-associated signal pathways, such as the PI3K/AKT pathway, and a suppression of oxidative stress via the HO-1/Nrf-2 system. GNL has also been reported to affect HO-1/Nrf-2 expression and has biological effects in different cell lines [52,58]. According to Western blot analysis, the HO-1,γ-GCLC and Nrf-2 protein expression levels of HUVECs treated with Ox-LDL were lower than those of cells treated with Ox-LDL and GNL (*p* < 0.05). The rate-limiting enzyme in heme catabolism is HO-1. When HO-1 is activated, it can exert cytoprotective, anti-inflammatory, anti-oxidative and inhibitory-apoptotic effects. There’s a link between modulating HO-1 and ROS production in other cell lines [59,60]. Therefore, upregulation of HO-1 via GNL could play a role in attenuating the production of intracellular ROS in GNL with Ox-LDL. GNL exposure significantly increased Nrf-2 expression (*p* < 0.05) compared to Ox-LDL incubation alone. These results show that GNL is involved in inhibiting oxidative stress by affecting the HO-1/Nrf-2 system.

The PI3K/Akt pathway is critical in promoting cell survival in vascular cells, and there’s increasing evidence of cross-talk between the Nrf2 and PI3K/Akt pathways in response to oxidative stress [17,61,62]. We hypothesized that PI3K/AKT regulates GNL in cells treated with Ox-LDL. AKT phosphorylation was dramatically increased (*p* < 0.05) by GNL supplementation, suggesting that GNL stimulates Nrf2 expression by phosphorylating AKT. LY294002, an inhibitor of the PI3K/AKT pathway, as well as GNL and Ox-LDL were used to assess the PI3K/AKT pathway’s function. GNL induced PI3K/AKT and Nrf2 expression in the study. The results show that PI3K/AKT signaling plays a crucial role in GNL-induced HO-1 expression by inducing Nrf2 in oxidative stress by Ox-LDL.

## 5. Conclusions

In this study, GNL showed therapeutic and prophylactic efficacy against Ox-LDL-associated oxidative endothelial damage. In addition, GNL inhibits Ox-LDL-stimulated inflammation by lowering the pro-inflammatory cytokines and downregulating ICAM-1 and VACM-1 expression, which means that GNL inhibits Ox-LDL-stimulated inflammation by lowering the pro-inflammatory cytokines. Moreover, in OX-LDL-stimulated endothelium, GNL inhibited ROS-mediated inflammation activation through Nrf2 activation, by inhibiting ROS-mediated inflammation activation. Through PI3K/AKT/Nrf2/HO-1, GNL suppressed Ox-LDL-mediated ROS production, improved the cell viability and blocked cell apoptosis. Further in-depth studies will aim to establish bioactive GNL as a potential candidate for treating oxidative stress-induced AS complications.

## Figures and Tables

**Figure 1 nutrients-14-04817-f001:**
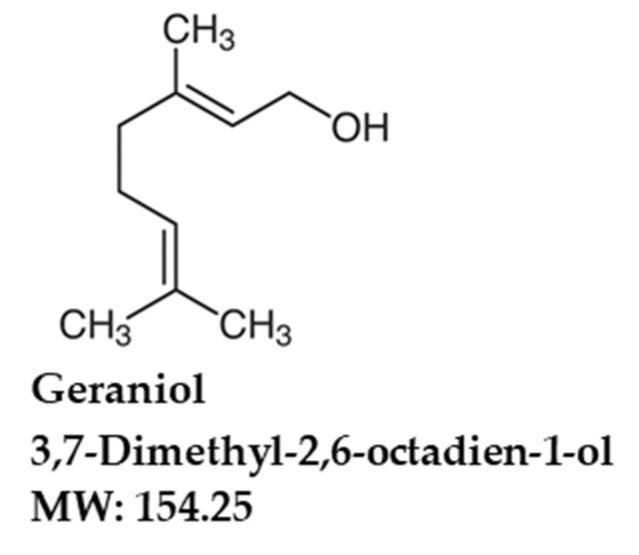
Chemical structure of geraniol (GNL).

**Figure 2 nutrients-14-04817-f002:**
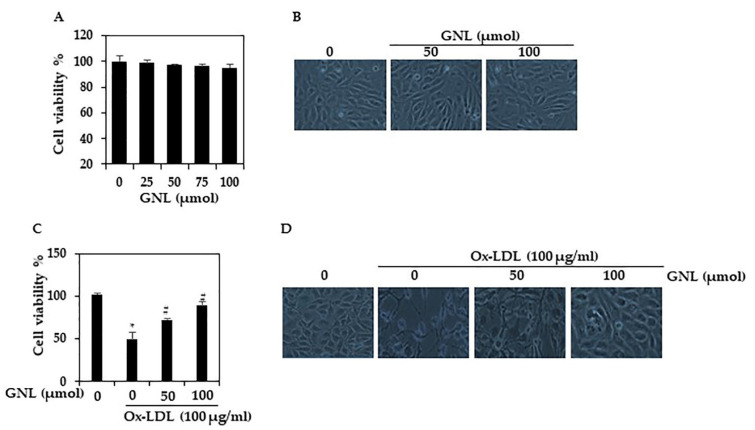
GNL chemical structure, its effects on Ox-LDL-induced cell viability, and the morphological changes. (**A**) HUVECs were added with GNL for 24 h and viability was measured by MTT. (**B**) GNL effect on the morphology of HUVECs. (**C**) In an MTT assay, GNL prevents Ox-LDL’s cytotoxicity (24 h). There are three replicates of each value, and * *p* < 0.05 represents a significant difference when compared to the control group. The # *p* < 0.05 represents significant differences between Ox-LDL alone and GNL with Ox-LDL treatment groups. (**D**) In Ox-LDL-induced HUVECs, GNL protects cell morphology.

**Figure 3 nutrients-14-04817-f003:**
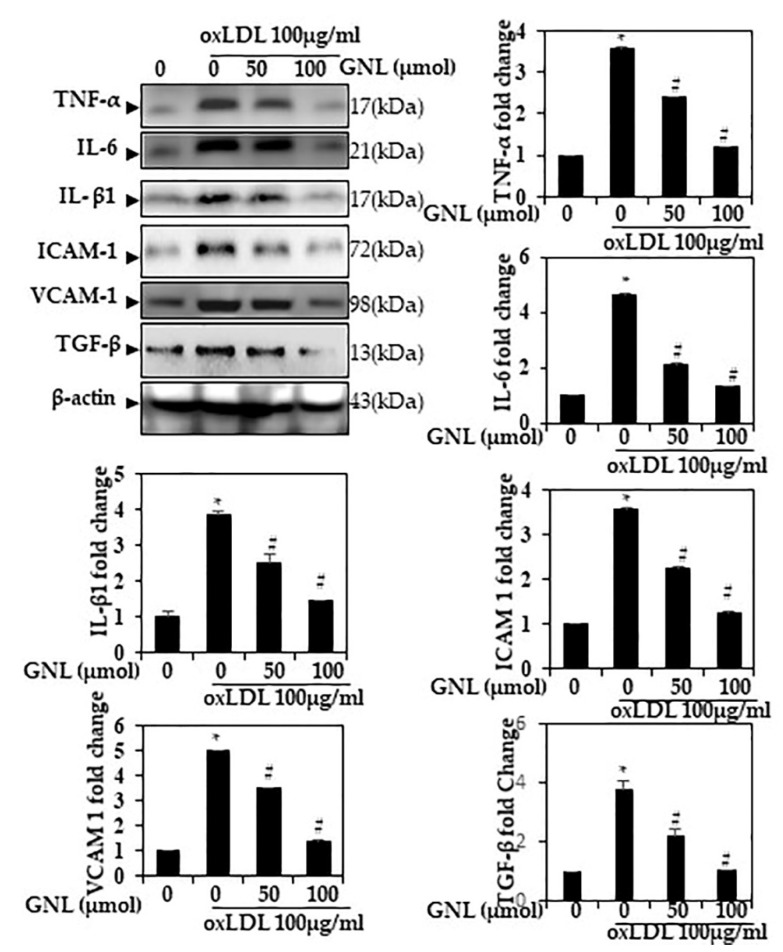
GNL mitigated Ox-LDL-induced cytokine production in endothelial cells. HUVECs were treated with GNL and/or Ox-LDL for 24 h. Expression of TNF-α, IL-6, IL-1β, ICAM-1, VCAM-1 and TGF-β protein was assessed by Western blotting. There are three replicates of each value, and * *p* < 0.05 represents a significant difference when compared to the control group. A # represents *p* < 0.05; thus, significant differences between the Ox-LDL alone and GNL with Ox-LDL treatment groups.

**Figure 4 nutrients-14-04817-f004:**
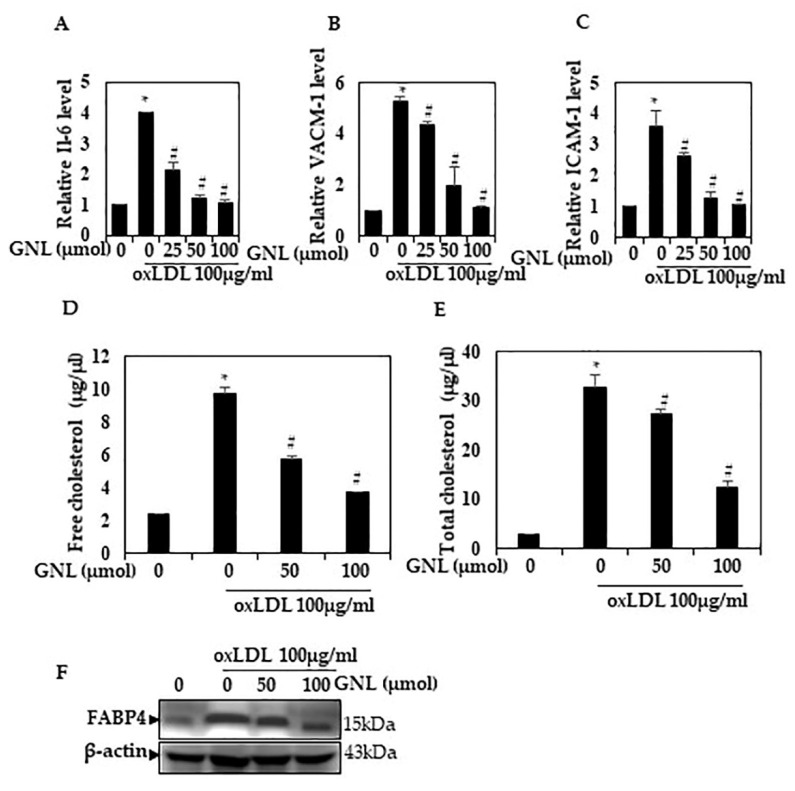
Effect of GNL on Ox-LDL-induced deposition of cholesterol in HUVECs. (**A**–**C**) RT-PCR quantification of pro-inflammation cytokines IL-6 and ICAM-1 and adhesion molecule ICAM-1. (**D**) Free cholesterol (FC); (**E**) Total cholesterol (TC). (**F**) Quantification of fatty acid-binding protein 4 (FABP4) activation by Western blot analyses. There are three replicates of each value, and * represents *p* < 0.05; thus, a significant difference when compared to the control group. The # represents *p* < 0.05; thus, significant differences between the Ox-LDL alone and GNL with Ox-LDL treatment groups.

**Figure 5 nutrients-14-04817-f005:**
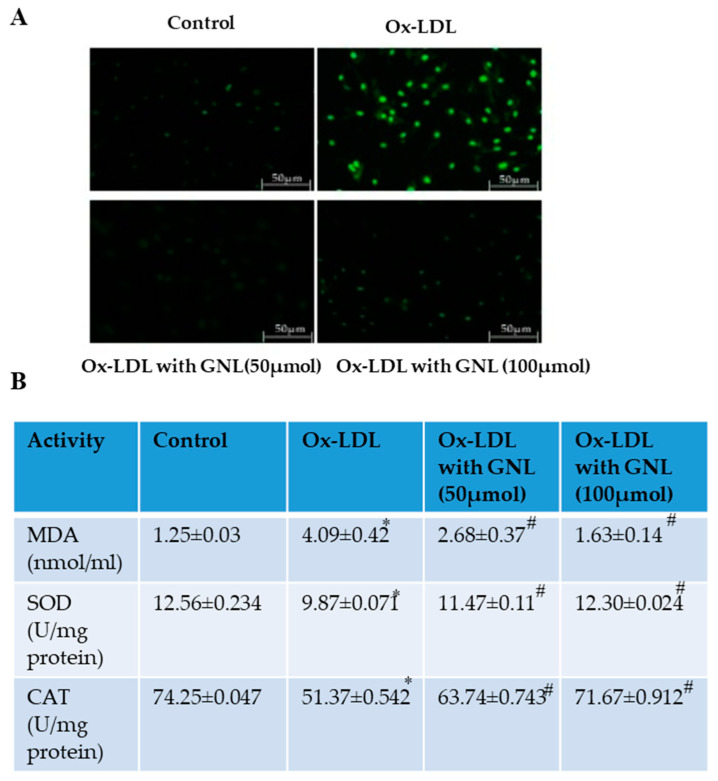
Effect of GNL on Ox-LDL-induced ROS production in HUVECs. (**A**) The HUVECs were then pretreated with GNL (0, 50 and 100 µM, for 2 h), followed by Ox-LDL (100 µg/mL) for 24 h. We measured the intracellular ROS levels using DCF fluorescence. (**B**) LPO, SOD and CAT. Based on the manufacturer’s instructions, we used ELISA kits. There are three replicates of each value, and * represents *p* < 0.05; thus, there is a significant difference when compared to the control group. The # represents *p* < 0.05; thus, there are significant differences between the Ox-LDL alone and GNL with Ox-LDL treatment groups.

**Figure 6 nutrients-14-04817-f006:**
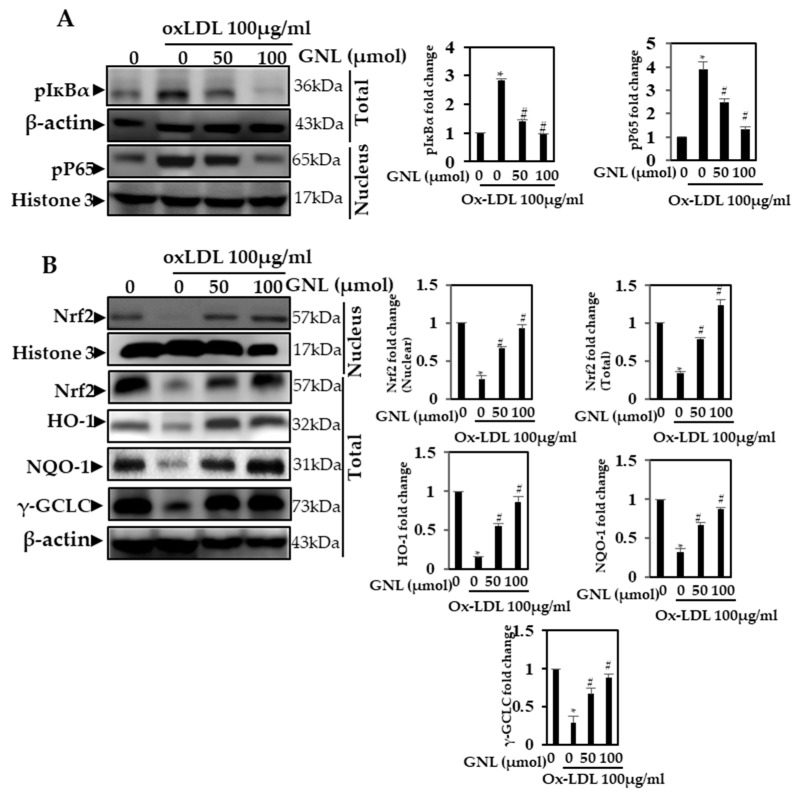
NF-ᴋB p65 expression is affected by GNL. (**A**) p-IᴋBα and pNF-ᴋB p65 antibodies were used to detect nuclear protein extract and total protein extract on 10–12% SDS-PAGE (polyacrylamide gel electrophoresis). (**B**) By Western blot analysis, we measured the levels of nuclear Nrf2 and NQO-1, HO-1 and γ-GCLC. There are three replicates of each value, and * represents *p* < 0.05; thus, there is a significant difference when compared to the control group. The # represents *p* < 0.05; thus, there are significant differences between the Ox-LDL alone and GNL with Ox-LDL treatment groups.

**Figure 7 nutrients-14-04817-f007:**
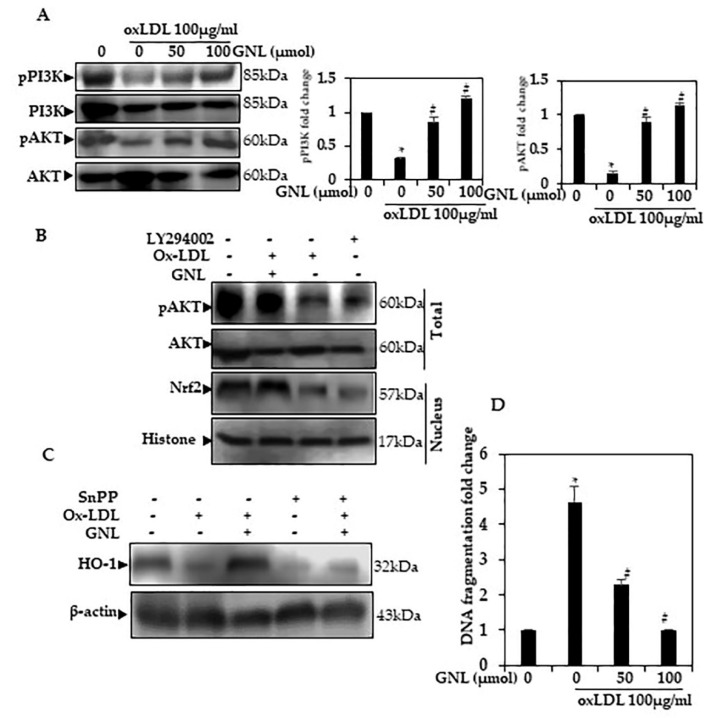
GNL activates PI3K/AKT phosphorylation. (**A**) The cells were treated with Ox-LDL (100 µg/mL) or GNL (0, 50 and 100 μmol) for 24 h. Based on Western blots, GNL has an impact on anti-pPI3K and anti-pAKT. Th relative ratios of PI3K and AKT expression are listed in the Western blotting results. (**B**) Cells were pretreated for two hours with PI3K/AKT inhibitors, and then with GNL (100 mol) and/or Ox-LDL (100 g/mL) for 24 h. By using anti-pAKT and anti-Nrf2, Western blotting showed pAKT, Nrf2 levels. (**C**) After being exposed to SnPP (40 µM/mL) for 1 h, cells were treated with Ox-LDL and GNL for 24 h, and proteins were analyzed by Western blot. (**D**) DNA fragmentation. There are three replicates of each value, and * represents *p* < 0.05; thus, a significant difference when compared to a control group. The # represents *p* < 0.05; thus, significant differences between the Ox-LDL alone and GNL with Ox-LDL treatment groups.

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
