# Peer review of "Anti-Inflammatory Activity of Geraniol Isolated from Lemon Grass on Ox-LDL-Stimulated Endothelial Cells by Upregulation of Heme Oxygenase-1 via PI3K/Akt and Nrf-2 Signaling Pathways"

_nutrients, 2022, doi:10.3390/nu14224817_

Round 1

Reviewer 1 Report

The work could be interesting, but unfortunately, it is written chaotically and contains numerous errors and mistakes. The authors did not check it carefully. This causes that in some places, it is not known what the meaning of the obtained results is and whether the presented results are true or whether it is an error.

The abstract should be redrafted. It is not clear which results are the authors' observations and which represent the state of knowledge from other publications. The abstract does not describe the work as required by the guidelines for authors: "The abstract should be a single paragraph and should follow the style of structured abstracts but without headings: 1) Background: Place the question addressed in a broad context and highlight the purpose of the study; 2) Methods: Describe briefly the main methods or treatments applied. Include any relevant preregistration numbers and species and strains of any animals used. 3) Results: Summarize the article's main findings; and 4) Conclusion: Indicate the main conclusions or interpretations. The abstract should be an objective representation of the article: it must not contain results which are not presented and substantiated in the main text and should not exaggerate the main conclusions."

The work has an incorrect order. According to the guidelines for authors, it should be: Introduction, Materials and Methods, Results, Discussion, and Conclusions.

Authors should organize the use of abbreviations. There is a rule that if it is introduced the first time it appears and only abbreviation is used in the further text (AS line 23, again, line 41). Meanwhile, the authors use different abbreviations several times, which is confusing (Geraniol (GNL) - line 23, Geraniol (GE), line 74). They also use different abbreviations for the same: CWDs / CWD; Ox-LDL / Ox - LDL / ox - LDL; HUVECs / HUVEcs / HUVEC; Western blot / western blot ...

There should be no abbreviations in the abstract because they make no sense there. We start to do it in the content of the article.

The authors gave the entire introduction to Cardiovascular diseases and atherosclerosis. However, there is no information on this in the abstract.

It is also worth changing the title of the work to explain the research area in which the authors expect benefits of GNL (AS, CVDs, ox-LDL?)

Figure 1 is chaotic. Figure 1A does not correspond to the description of Figure 1. It is difficult to find where the letters A-E are. A better layout would be if D were placed below B and E below C. The statistical description under Figure 1 is for the D-plot only, so it is unknown why it is placed at the end of all subfigures (A-E)? It is not clear what the # sign refers to - we always compare the test group to the control group. The description shows that the significance is the opposite - ox-LDL 0 to ox-LDL100 + GNL50 and vs ox-LDL100 + GNL100. Meanwhile, the Figure shows that only ox-LDL100 + GNL100 vs ox-LDL 0 is significant.

Figure 1A does not match the Results chapter. The chemical formula of compounds is either for Introduction or Materials and Methods.

In chapter 2.1 there is no information about the statistical significance of the obtained results - they should be found here primarily, in Figure 1 they are only an explanation of the Figure, and they are unclear.

Figure 2 is chaotic. There is no information that this is an ELISA test. In turn, in chapter 2.2. there is no information that a WB test was carried out. What does "fold change" mean? If the test was performed by ELISA with a concentration curve, the authors should show the absolute values ​​together with the units.

Description: "There are three replicates of each value, and * p<0.05 represents a significant difference when compared to a control group. The #p<0.05 represents significant differences between Ox-LDL alone and GNL with Ox-LDL treatment groups" is pasted into each Figure (1,2,3,4,5,6) but does it concern all the presented studies? MTT, WB, ELISA, PCR ...? The legend should be relevant to the content, not universal. It is confusing to portray materialities that are either missing or untrue.

Again, the description: "#p<0.05 represents significant differences between Ox-LDL alone and GNL with Ox-LDL treatment groups" is in the wrong order and suggests that # denotes significance for both GNL groups with Ox-LDL while in the figures, it is only for GNL100 + Ox-LDL100.

Did the ELISA also apply to VCAM-1 and ICAM-1? They are not listed in chapter 2.2. They are listed in chapter 2.3 but for WB, what then are the plots of VCAM-1 and ICAM-1 in figure 2? The ELISA test is not described at all in Figure 2.

The authors again in chapters 2.1 - 2.15 do not provide information on the statistical significance of the results. If the information appears, it is not clear to which group it applies exactly.

Chapter 2.4 - the authors describe the results: "qRT-PCR showed significant downregulation of TNF-α, VCAM-1, and ICAM-1 mRNA" while the Figure shows: qRT-PCR for IL-6, TNF-α, ICAM-1. In the description, however, there is again information: "qRT-PCR showed significant downregulation of TNF-α, VCAM-1, and ICAM-1 mRNA". ??????

Here, again the authors describe the statistical significance for the studied parameters, but it is not known what groups and concentrations they apply to.

Line 154, lack of references

In figure 4B, the significances again appear to be incorrectly marked. The ox-LDL + GRL100 group has higher values than the GNL50 relative to the control and yet the results are not significant. Please verify. The statistical significance description is again copy & paste, it is not specific to this Figure.

The authors use the phrase everywhere:"were observed in the groups that received Ox-LDL in comparison with the control group (p < 0.05)" What is very unclear as the results are for two GNL concentrations and the authors do not distinguish which group is meant, the more that significance is marked for only one of them.

Chapter 2.9, line 189. There is no information given there. The description is likely to refer to Figure 4b rather than 2b.

Figure 5 - Same comments as for Figure 2. The authors quote the number of times without giving absolute measured values, no units are given. The Figure is illegible, the letters are blurry and too small. Are the graphs taken from ELISA? is not explained. The statistical significance description is again copy & paste, it is not specific to this Figure.

Chapter 2.11, line 213. The description is incorrect. it probably concerns figure 6a rather than 5a.

Line 226, is the information provided actually shown in figure 6A? The results report does not provide information about: Nrf2, HO-1, NQO-1, and γ-GCLC

The remarks for Figure 6A are the same as for the previous ones.

Reviewer 2 Report

The authors presented a study entitled "Anti-inflammatory activity of geraniol isolated from lemon grass on Ox-LDL-stimulated endothelial cells by upregulation of heme oxygenase-1 via PI3K / Akt and Nrf-2 signaling pathways". There is good data to support a scientific discussion in the manuscript. I hope readers will enjoy reading this article. This is an interesting study in which methods and experiments are documented in detail. The results are presented appropriately, clearly explaining the study results. The manuscript is written in a direct and active style. Overall, the paper offers sufficient detail on the methodology and is written comprehensively enough to be understandable.

In my opinion, some small remarks should be addressed:

- in line 40-41 a note is needed.

- line 74 the geraniol is called GE but after it is named GNL, standardize the manuscript.

- I advise the authors to broaden the discussions, in light of the interesting results obtained in the experimental section.

- for the same reason the conclusions are really not representative of the work done, the authors could contextualize the results, it could be indicated how this study could influence the clinic of the pathology or what other studies would be needed in the future for a better understanding.

Author Response

Responses to Reviewer 2 comments

Thank you for giving us the opportunity to submit a revised draft of our manuscript entitled “Anti-inflammatory activity of geraniol isolated from lemon grass on Ox-LDL-stimulated endothelial cells by upregulation of heme oxygenase-1 via PI3K/Akt and Nrf-2 signaling pathways” [Manuscript ID: nutreints 1994945] to Nutreints.

We appreciate the time and effort that you and the reviewers have dedicated in providing valuable feedback on our manuscript. We are grateful to the reviewers for their insightful comments, which have improved the paper.

  • Most of the work carried out shows an anti-inflammatory effect of geraniol on endothelial cells stimulated by Ox-LDL. The authors chose as inflammatory markers mainly the inflammatory cytokines (TNF-alpha, IL-1beta and IL-6) and the transcription factor NF-kB. As cyclooxygenase-2 (COX-2) is regulated by all these factors and is itself involved in inflammatory processes through the synthesis of, among others, prostaglandin E2 (PGE2), it is absolutely necessary that the analysis of the enzymatic pathway is carried out both by looking for the expression of COX-2 and by measuring PGE2.

Thank you for your valuable comments. Unfortunately, there was no available anti-iNOS, anti-Cox-2, and anti-PGE2 antibody. These antibodies were ordered; our suppliers took more than 5 months to deliver them to us.

We have considered your valuable suggestion for future animal studies.

  • The analysis of the expression of cytokines is not sufficient, it is also necessary to carry out the assay of cytokines such as TNF-alpha, IL-1beta and IL-6.

We have done this experiment according to your suggestion, in the supplementary file 1.

  • A μmol unit for GNL means nothing, I assume it is μM so it has to be changed in the whole manuscript and especially on figures.

We have made the correction as per your suggestion.

4) For IkBalpha, the authors show the increase in the expression of the native
form but it is essential to show also the expression of its phosphorylated form which should be either stable or decreased in correlation with
sequestration of NF-kB in the cytoplasm.

Thank you for your comments.

Ikkb blot actually replicate of Nrf2, by mistake we put IкBα, sorry for this mistaken.

We have corrected this in our revised manuscript (IкBα WB result).

Thank you again for your valuable comments and suggestions that will certainly improve the quality of our manuscript.

 If any responses are unclear or you wish additional changes, please do not hesitate to let us know

Reviewer 3 Report

Major Points

1) Most of the work carried out shows an anti-inflammatory effect of geraniol on endothelial cells stimulated by Ox-LDL. The authors chose as inflammatory markers mainly the inflammatory cytokines (TNF-alpha, IL-1beta and IL-6) and the transcription factor NF-kB. As cyclooxygenase-2 (COX-2) is regulated by all these factors and is itself involved in inflammatory processes through the synthesis of, among others, prostaglandin E2 (PGE2), it is absolutely necessary that the analysis of the enzymatic pathway is carried out both by looking for the expression of COX-2 and by measuring PGE2.

2) The analysis of the expression of cytokines is not sufficient, it is also necessary to carry out the assay of cytokines such as TNF-alpha, IL-1beta and IL-6.

3) A μmol unit for GNL means nothing, I assume it is μM so it has to be changed in the whole manuscript and especially on figures.

4) For IkBalpha, the authors show the increase in the expression of the native
form but it is essential to show also the expression of its phosphorylated form which should be either stable or decreased in correlation with
sequestration of NF-kB in the cytoplasm.

Author Response

Responses to Reviewer 3 comments:

The authors presented a study entitled "Anti-inflammatory activity of geraniol isolated from lemon grass on Ox-LDL-stimulated endothelial cells by upregulation of heme oxygenase-1 via PI3K / Akt and Nrf-2 signaling pathways". There is good data to support a scientific discussion in the manuscript. I hope readers will enjoy reading this article. This is an interesting study in which methods and experiments are documented in detail. The results are presented appropriately, clearly explaining the study results. The manuscript is written in a direct and active style. Overall, the paper offers sufficient detail on the methodology and is written comprehensively enough to be understandable.

Thank you for giving us the opportunity to submit a revised draft of our manuscript entitled “Anti-inflammatory activity of geraniol isolated from lemon grass on Ox-LDL-stimulated endothelial cells by upregulation of heme oxygenase-1 via PI3K/Akt and Nrf-2 signaling pathways” [Manuscript ID: nutreints 1994945] to Nutreints.

We appreciate the time and effort that you and the reviewers have dedicated in providing valuable feedback on our manuscript. We are grateful to the reviewers for their insightful comments, which have improved the paper.

In my opinion, some small remarks should be addressed:

- in line 40-41 a note is needed.

Thank you for your valuable comments. As suggested, in our revised manuscript we have corrected it.

- line 74 the geraniol is called GE but after it is named GNL, standardize the manuscript.

As suggested, in our revised manuscript we have corrected it.

- I advise the authors to broaden the discussions, in light of the interesting results obtained in the experimental section.

As suggested, in our revised manuscript we have corrected it.

- for the same reason the conclusions are really not representative of the work done, the authors could contextualize the results, it could be indicated how this study could influence the clinic of the pathology or what other studies would be needed in the future for a better understanding.

As suggested, in our revised manuscript we have modified our conclusion part.

Thank you again for your valuable comments and suggestions that will certainly improve the quality of our manuscript.

 If any responses are unclear or you wish additional changes, please do not hesitate to let us know

Round 2

Reviewer 1 Report

line 82 - 94 Geraniol formula should be described as Figure 1. and cited in the text.

Figure 2. GNL chemical structure should be removed - it's not on this Figure. Increase the statistical significance's size/resolution (?) on all figures.

Figure 6. Increase the size of the pictures on this figure. They are smaller than others and illegible.

Author Response

Reviewer 2 V2

Thank you for your comments

line 82 - 94 Geraniol formula should be described as Figure 1. and cited in the text.

 Thank you for your comment. As per your suggestion, Geraniol formula has been described as Figure 1 and cited in the text.  (Line number 74)

Figure 2. GNL chemical structure should be removed - it's not on this Figure. Increase the statistical significance's size/resolution (?) on all figures.

 As suggested have done our revised manuscript

Figure 6. Increase the size of the pictures on this figure. They are smaller than others and illegible.

Every image has been resized and resized to 300 dpi in order to improve quality and size
